# Comparative Transcriptome Analysis of Two *Aegilops tauschii* with Contrasting Drought Tolerance by RNA-Seq

**DOI:** 10.3390/ijms21103595

**Published:** 2020-05-19

**Authors:** Xinpeng Zhao, Shenglong Bai, Lechen Li, Xue Han, Jiahui Li, Yumeng Zhu, Yuan Fang, Dale Zhang, Suoping Li

**Affiliations:** 1Key Laboratory of Plant Stress Biology, State Key Laboratory of Cotton Biology, School of Life Sciences, Henan University, Kaifeng 475001, China; xinpeng_henu@163.com (X.Z.); slbai@vip.henu.edu.cn (S.B.); lilechen1234@126.com (L.L.); hanxue0727@163.com (X.H.); lijiahui2756@163.com (J.L.); zhuyumeng2023@163.com (Y.Z.); lisuoping@henu.edu.cn (S.L.); 2College of Agriculture, Nanjing Agricultural University, Nanjing 210095, China; hedafangyuan@163.com

**Keywords:** coleoptile length, differentially expressed genes, rate of water loss, physiological traits, RNA sequencing

## Abstract

As the diploid progenitor of common wheat, *Aegilops tauschii* is considered to be a valuable resistance source to various biotic and abiotic stresses. However, little has been reported concerning the molecular mechanism of drought tolerance in *Ae. tauschii*. In this work, the drought tolerance of 155 *Ae. tauschii* accessions was firstly screened on the basis of their coleoptile lengths under simulated drought stress. Subsequently, two accessions (XJ002 and XJ098) with contrasting coleoptile lengths were selected and intensively analyzed on rate of water loss (RWL) as well as physiological characters, confirming the difference in drought tolerance at the seedling stage. Further, RNA-seq was utilized for global transcriptome profiling of the two accessions seedling leaves under drought stress conditions. A total of 6969 differentially expressed genes (DEGs) associated with drought tolerance were identified, and their functional annotations demonstrated that the stress response was mediated by pathways involving alpha-linolenic acid metabolism, starch and sucrose metabolism, peroxisome, mitogen-activated protein kinase (MAPK) signaling, carbon fixation in photosynthetic organisms, and glycerophospholipid metabolism. In addition, DEGs with obvious differences between the two accessions were intensively analyzed, indicating that the expression level of DEGs was basically in alignment with the physiological changes of *Ae. tauschii* under drought stress. The results not only shed fundamental light on the regulatory process of drought tolerance in *Ae. tauschii*, but also provide a new gene resource for improving the drought tolerance of common wheat.

## 1. Introduction

Common wheat, owing to its high yield, nutritional and processing qualities, has been one of the most widely cultivated crops worldwide, and accounts for approximately 20% of consumed calories by humans [1]. It should be noted that the genetic background of common wheat is becoming narrower through the process of domestication, selection, and modern breeding, leading to the gradual decline in the ability to resist biotic and abiotic stresses [2,3].

Drought is one of the most prevalent abiotic stresses around the world, and causes a serious decrease in crop production and quality [4]. In this regard, the development of a wheat variety with drought tolerance is an effective approach to improving and maintaining productivity of wheat. However, limited by their narrow genetic background, it has become increasingly difficult to obtain a fine wheat variety with drought tolerance and high yield by way of hybridization among varieties. Fortunately, a greater tolerance to drought has been found in their wild relatives compared with domesticated descendants, which provides new genes and alleles to improve the drought tolerance and enrich the genetic background of common wheat [5].

*Aegilops tauschii* Cosson (DD, 2n = 2x = 14), the diploid progenitor of common wheat, is found naturally distributed in central Eurasia, spreading from northern Syria and Turkey to western China (Yili area of Xinjiang) [6]. It has adapted to diverse environments including margins of deserts, sandy seashore, steppes, stony hills, roadsides, wastelands and humid temperate forests [7], thus forming abundant genetic diversity. On the basis of its genetic background, it is preferable to subdivide *Ae. tauschii* into two phylogenetic lineages, designated as L1 and L2. The former is broadly related to *Ae. tauschii* ssp. *tauschii,* and the latter is generally correlated with *Ae. tauschii* ssp. *strangulata* [8,9]. It is widely accepted that *Ae. tauschii* (mainly L2 lineage) from Transcaucasus and northern Iran is involved in the origin of the wheat D genome. Owing to the long genetic distance between L1 and L2, *Ae. tauschii* (mainly L1 lineage) from the eastern and southern populations (i.e., those from Syria, Afghanistan, Pakistan, Central Asia, and China) has more abundant genetic and phenotypic characteristics [6]. Therefore, like many wild crop progenitors, *Ae. tauschii* is considered to be a valuable resistance source to various biotic and abiotic stresses [10].

Far from a simple trait controlled by a single gene, drought tolerance is mostly conditioned by many component responses [11]. Under drought stress, a series of cellular responses are adopted by plants, including morphological, physiological and biochemical processes, such as coleoptile growth, stomatal closure, membrane stability change, osmoprotectants and antioxidants accumulation, reactive oxygen species (ROS) scavenging and transcription activation [12]. All of these responses are controlled by regulating the expression of various genes at the molecular level. Previous studies have demonstrated considerable genetic variability in coleoptile length response to low water potentials at the very early seedling stage in wheat [13,14]. The positive relationship between coleoptile growth and grain yield under drought stress has also been demonstrated in wheat [15]. Therefore, coleoptile length was considered to be a good parameter to distinguish drought-resistant types from drought sensitive wheat in the early seedling stage.

In recent years, transcriptome analysis of stress tolerance has been applied in common wheat and its wild relative species using microarrays [16,17] and RNA-Seq [18,19,20], and has helped to reveal the molecular mechanisms and discover many candidate genes for various types of abiotic stress tolerance. Mansouri et al. [21] investigated the transcriptional changes of *Ae. tauschii* leaves associated with long-term salt stress utilizing paired-end sequencing technology. All of the generated unigene sequences were aligned to public protein databases, such as the National Center for Biotechnology Information (NCBI) nonredundant (nr) protein database and SwissProt, to obtain the annotation of differentially expressed unigenes (DEUs) since no draft genome sequence of *Ae. tauschii* with high quality could be referred. Almost at the same time, the whole genome fine mapping of *Ae. tauschii* was established using next-generation sequencing [22,23], providing a reliable reference genome for transcriptome analysis of *Ae. tauschii*. The high quality reference genome is believed to nicely illustrate the complex gene regulatory networks acquired by RNA-seq. To gain a comprehensive understanding of the molecular mechanism involved in response to drought stress at the seedling stage of *Ae. tauschii*, we screened the drought tolerance of 155 *Ae. tauschii* accessions on the basis of their coleoptile lengths under simulated drought stress. Subsequently, two accessions (XJ002 and XJ098) with contrasting coleoptile lengths were selected and intensively analyzed on their rate of water loss (RWL) as well as physiological characters to confirm their difference of drought tolerance. RNA-seq was used to identify their drought-responsive differentially expressed genes (DEGs) involved in metabolic pathways under control and drought stress. The identification of DEGs in this species may provide new wild gene resource for improving drought tolerance of common wheat.

## 2. Result

### 2.1. Coleoptile Length of Ae. tauschii under Simulated Drought Stress

Under both drought stress and control treatments, the mean coleoptile lengths (CLs) of 155 *Ae. tauschii* accessions were determined to be 0.10–0.84 cm and 0.26–2.4 cm, with C.V values of 0.46 and 0.36, respectively. This indicates the huge difference among the *Ae. tauschii* accessions in drought tolerance (Appendix A). Further, the Euclidean distances were calculated on the basis of CL values under drought stress and control treatments as well as their ratios, which fall into the range of 0.21–6.11 with an average of 2.08. All 155 accessions were further clustered through the method of unweighted pair-group method with arithmetic means (UPGMA) according to their Euclidean distances (Appendix A). As could be observed in this figure, *Ae. tauschii* can be divided into two groups at the Euclidean distance 2.62. The *t*-test of the above mentioned three parameters shows an extremely significant difference (*p* < 0.01) between the two groups. This result confirms the wide difference in the ability of *Ae. tauschii* from the Xinjiang area and Yellow River basin against drought stress.

### 2.2. Water Loss Rate and Physiological Characters of Two Ae. tauschii with Contrasting Drought Tolerance

Two accessions (XJ002 and XJ098) with contrasting coleoptile lengths were selected and intensively analyzed to comprehend the variation under drought stress. At the *Ae. tauschii* seedling stage, the rate of water loss (RWL) between the two accessions showed a significant difference (Figure 1, Appendix A). Compared with XJ002, XJ098 had a higher RWL, further confirming its poor drought resistance.

The physiological traits of *Ae. tauschii* seedlings, including proline contents, malonaldehyde (MDA), water soluble sugar (WSS), peroxidase (POD), polyphenol oxidase (PPO), and relative electrolyte leakage, were obtained under control and 20% PEG-6000 to simulate drought stress treatments (Figure 2, Appendix A). As displayed in Figure 2, the physiological traits of XJ002 and XJ098 displayed few changes in the control condition. While under the drought stress treatment, the proline content of XJ002 and XJ098 exhibited continuous accumulation (Figure 2A), which helps in maintaining water content and turgor pressure in plant cells to enhance drought tolerance at the early stage of stress [24]. The proline content of XJ002 was 1.61-fold (*p* < 0.05) higher compared with that of XJ098 at the 10 h. The WSS content is believed to be positively correlated with drought stress [25], and keeps slowly increasing in the initial 5 h for both XJ002 and XJ098, followed by a remarkable difference at 8 h for the WSS content (Figure 2B). In this case, the WSS content reached 1.40-fold (*p* < 0.05) higher in XJ002 than that in XJ098.

It is widely accepted that POD plays critical roles in eliminating MDA, decreasing H_2_O_2_ accumulation, maintaining cell membrane integrity and resisting peroxidation of membrane lipids [25]. The POD activities of XJ002 and XJ098 sharply increased after treatment for 8 h, and both reached their peak values at 10 h under the stress condition (Figure 2C). In this case, the POD activity of XJ002 is 2.46-fold (*p* < 0.05) higher than that of XJ098. Meanwhile, a similar tendency could be observed for PPO and POD activities in the process of drought stress. At 10 h, the PPO activity of XJ002 is 1.61-fold (*p* < 0.05) that of XJ098 (Figure 2D). Both of the PPO and POD activities remarkably decrease along with continuous drought stress.

As the final product of lipid peroxidation, MDA is often applied as an indicator of cell oxidative damage under abiotic stress [26]. The MDA content of XJ002 and XJ098 dramatically increased after stress treatment for 5 h due to accumulation of lipid peroxidation. Overall, the MDA content of XJ098 increased more sharply than that of XJ002, and the most prominent difference could be observed at 8 h, with 1.42- fold (*p* < 0.05) higher MDA content in XJ098 than that in XJ002 (Figure 2E). The changes of membrane permeability under drought stress were evaluated by the relative electrolyte leakage [27], and was continuously monitored after 5 h of treatment along with the deepening of drought stress in this work. After treatment for 20 h, the membrane electrolyte leakage of XJ002 increased to 1.17-fold compared with that of XJ098 (Figure 2F). In summary, based on the above analysis of physiological traits, XJ002 exhibits stronger drought tolerance than XJ098 at the seeding stage.

### 2.3. RNA-Seq Analysis of Two Ae. tauschii with Contrasting Drought Tolerance

To better understand the molecular mechanism of drought stress in *Ae. tauschii*, the total RNA of seedling leaves from the two above-mentioned accessions (XJ002 and XJ098) was sequenced using an Illumina system under control and drought stress conditions. To profile the *Ae. tauschii* response to drought stress, transcriptomic data of 12 samples were obtained in total, with three biological replications for each condition. The RNA-seq analysis provided 41.8–59.4 million raw reads per biological replicate with an average read length of 101 bp (Table 1). After filtering out adapter and low-quality reads, approximately 494.6 million clean reads were obtained in the 12 transcriptome libraries, in which Q20 percentages of each sample (sequencing error rates lower than 1%) were found to be higher than 96.97%. Meanwhile, 92.07–93.38% of clean reads could be well aligned with the *Ae. tauschii* reference genome in total [19], 89.47–90.11% could be accurately mapped to a specific location within the reference genome sequence, and 2.38–3.69% could be mapped to multiple locations. The gene expression level could be quantified by normalized FPKM [28]. Therefore, the FPKM data were tested to evaluate correlations among biological replicates, and all the obtained Pearson correlation coefficients among biological replicates were found to be higher than 0.96 (Appendix A). In addition, principal component analysis (PCA) was performed to visualize the variation among samples utilizing DESeq2 [29], and the biological replications for each sample were well clustered together (Appendix A). Overall, the transcriptomic data in this work were viable on the basis of statistical analysis in *Ae. tauschii* RNA sequencing.

### 2.4. Identification and Analysis of DEGs

The two *Ae. tauschii* accessions were compared under the control and stress conditions to identify DEGs (*q*-value ≤ 0.05 and |log2 FC| ≥ 1) by DESeq2 software. A total of 5401 genes were differentially expressed in the draught-tolerant genotype (XJ002), with 2429 up-regulated and 2972 down-regulated. In comparison, 5910 DEGs were identified in the drought-sensitive genotype (XJ098), with 2818 up-regulated and 3092 down-regulated under drought stress conditions (Figure 3A). All of the DEGs in XJ002 and XJ098 genotypes are listed in Appendix A, respectively. By comparing the drought stress responses between the two accessions at the gene level, 4336 DEGs were found to be consistent under drought stress, including 2055 up-regulated and 2281 down-regulated genes (Figure 3B). Furthermore, the regulation patterns of the six genes (*AET4Gv20722600*, *AET5Gv20147500*, *AET5Gv20023600*, *AET7Gv20860400*, *AET4Gv20013900*, and *AET2Gv20043100*) between the two genotypes were revealed to be adverse. Based on hierarchical clustering analysis of gene expression, a total of 6969 DEGs in the two *Ae. tauschii* accessions could be roughly grouped into seven classes (Figure 4). The relative expression level of most DEGs was found to be similar between the two genotypes, besides obvious differences in a few ones.

### 2.5. GO and KEGG Enrichment Analysis of DEGs

To determine the fundamental functions of the obtained DEGs, GO analysis was performed using ClusterProfile R packages [30]. A total of 3547 and 3861 DEGs were respectively enriched in 83 GO terms from XJ002 (drought tolerance) (Appendix A) and 52 GO terms from XJ098 (drought sensitivity) (Appendix A), and could be classified into three categories: biological process, molecular function, and cellular component. In particular, the GO terms (GO: 0006629) in XJ002 and XJ098 enriched DEGs higher than 100, which could be indexed to lipid metabolic process. The gene ontologies related to drought stress could also be observed, including ‘dioxygenase activity’, ‘glucosyltransferase activity’, and ‘calcium ion binding’. Further, the enriched GO terms from XJ002 and XJ098 were compared on the basis of their biological process (Figure 5). As a result, those related to ‘cellular carbohydrate metabolic process (GO: 0044262)’ and ‘cell redox homeostasis (GO: 0045454)’ were significantly enriched in the tolerant genotype. While in the sensitive genotype, gene ontologies correlated to ‘regulation of developmental process (GO: 0050793)’ were observed.

To identify the metabolic pathways of DEGs in response to drought stress, we firstly searched the DEGs against KEGG using KofamKOALA. A total of 1501 and 1623 DEGs were anchored in KO terms from XJ002 (drought tolerance) and XJ098 (drought sensitivity), respectively, wherein 707 and 212 DEGs were found prominently enriched in 18 pathways from XJ002 and 7 pathways from XJ098 (Figure 6, Appendix A). It is noteworthy that two pathways, ‘alpha-linolenic acid metabolism (ko00592)’ and ‘MAPK signaling pathway (ko04016)’ were found enriched in both of the two *Ae. tauschii.* Furthermore, three pathways were detected in the drought-tolerant genotype, including ‘starch and sucrose metabolism (ko00500)’, ‘peroxisome (ko04146)’, and ‘carbon fixation in photosynthetic organisms (ko00710)’. The pathway, ‘glycerophospholipid metabolism (ko00564)’, could only be obtained in the drought-sensitive genotype. From the above results, it could be established that the enriched GO terms and KEGG pathways were highly related with drought stress.

### 2.6. DEGs with an Obvious Difference between the Two Ae. tauschii

The DEGs enriched in GO terms and KEGG pathways were intensively compared between the two *Ae. tauschii* with contrasting drought tolerance. Given that most DEGs displayed a similar expression level under the control and drought stress conditions (Figure 4), DEGs with considerable difference between the two *Ae. tauschii* were acquired in the KEGG pathways related to drought stress. The *AET5Gv20023600* encoding lipoxygenase, which belongs to the ‘alpha-linolenic acid metabolism (ko00592)’ pathway, was observed to be down-regulated (log2FC 1.71↓) in the tolerant genotype (XJ002), but up-regulated (log2FC 1.20↑) in the sensitive genotype (XJ098). Only up-regulation could be found in XJ002 (log2FC 1.04↑) for the *AET5Gv20478000* (encoding trehalose-phosphatase), which was enriched in the ‘starch and sucrose metabolism (ko00500)’ pathway. Similarly, the *AET6Gv20051600* encoding sucrose 1-fructosyltransferase was observed to be more obviously up-regulated in XJ002 (log2FC 5.52↑) compared with that in XJ098 (log2FC 3.71↑). The *AET1Gv20883700* encoding mitogen-activated protein kinase kinase kinase (MAPKKK), belonging to the ‘MAPK signaling pathway (ko04016)’ pathway, was established more apt to be up-regulated in XJ002 (log2FC 4.51↑) compared with that in XJ098 (log2FC 2.16↑). Moreover, the *AET2Gv20156600* encoding peroxidase exhibited a more explicit tendency of up-regulation in XJ002 (log2FC 2.02↑), while little discrepancy could be found in the XJ098.

In addition, the DEGs encoding WRKY and NAC families between the two accessions exhibited a more significant difference compared with those of the other transcription factors (TFs). The *AET1Gv20300200* encoding WRKY24 was found to be obviously down-regulated in XJ002 (log2FC 1.42↓) with little difference in XJ098. While the *AET3Gv20244700* encoding WRKY24 showed more prominent up-regulation in XJ098 (log2FC 1.82↑) with little difference in the other species. For the *AET1Gv20866400* encoding WRKY51, more down-regulation could be observed in the tolerance genotype XJ002 (log2FC 4.15↓) than the sensitive genotype XJ098 (log2FC 2.71↓). Similarly, the *AET7Gv20860400* encoding WRKY70 was detected to undergo down-regulation in tolerant genotype XJ002 (log2FC 2.76↓), with an adverse tendency in sensitive genotype XJ098 (log2FC 1.96↑). Moreover, the *AET7Gv20524100* encoding NAC was observed to be up-regulated in XJ002 (log2FC 2.70↑) with little difference in XJ098.

### 2.7. Validation of RNA-Seq Analysis by Quantitative Real-Time PCR (qRT-PCR)

To determine the reliability of DEGs obtained from RNA-seq analysis in *Ae. tauschii* seedling leaves, 13 of them were randomly selected from the control and stress samples for qRT-PCR analysis (Appendix A), wherein *actin* was used as the reference gene for normalization. As a result, similar expression characteristics could be observed between qRT-PCR and RNA-Seq. The ratios of expression levels were further compared (Appendix A), revealing a significant correlation between the two methods (R^2^ = 0.8372, *n* = 26). This result unambiguously confirmed the reliability of the DEGs obtained from RNA-seq analysis in this study.

## 3. Discussion

As one of the most important environmental stresses in plants, drought seriously limits the production and quality of crops [31]. Breeding of drought-tolerant crops is highly valued to maintain yield in arid land. Wild relatives may provide new gene resources for improving drought tolerance of crops. RNA-seq is a useful approach for the identification of DEGs in a regulatory network at the transcriptome level, which provides insights into the molecular mechanisms in response to abiotic stresses [32]. Through the transcriptomic analysis of the maize seedling leaves under salinity, drought, heat, and cold stresses, the responses were found to be mediated by pathways involving hormone metabolism and signaling, transcription factors (TFs), very-long-chain fatty acid biosynthesis and lipid signaling [33]. Mansouri et al. [21] investigated transcriptome changes of *Ae. tauschii* leaf in long-term salt stress. They identified 4506 salt stress-responsive unigenes in total through differential expression analysis, which were found to be involved in ion homeostasis, signaling processes, carbohydrate metabolism, and post-translational modifications. In this work, a total of 6969 active DEGs were obtained by transcriptomic analysis in the *Ae. tauschii* seedling leaves under drought stress. Besides DEGs with remarkably higher number than that reported by Mansouri under salt stresses, more specific functional annotations of DEGs participating in the response to drought stress were established in this work, including phytohormone metabolism and signaling, transcriptional regulation and lipid signaling, owing to the very recently established whole genome fine mapping of *Ae. tauschii* [22,23]. It should be noted that finer controls would contribute to better discrimination between DEGs responsive to drought and other causes.

### 3.1. Peroxidases and Osmotic Regulation

In this study, a series of physiological and biochemical parameters of *Ae. tauschii* seedlings were analyzed under the 20% PEG-6000 for simulated drought stress, including proline contents, MDA, WSS, POD, PPO, and relative electrolyte leakage. The results indicated that XJ002 was more tolerant to drought than XJ098. Drought stress was found to be accompanied by the formation of ROS in different cellular compartments, which damages membranes and macromolecules [34]. Antioxidant enzymes (POD, PPO) and glutathione S-transferases (GSTs) could rapidly eliminate the ROS to minimize oxidative damage [35]. In this work, the two enzymes were activated in both genotypes after PEG treatment. Compared with those of XJ098, the POD and PPO activities of XJ002 were higher after PEG treatment (Figure 2C,D). The POD, PPO, and glutathione S-transferases (GSTs) were unexceptionally activated in response to drought stress. At the transcriptional level, the DEGs (*AET1Gv20485200*, *AET2Gv20156600*) encoding POD and GSTs were shown to be more obviously up-regulated in XJ002 (log2FC 4.64↑ for the former and log2FC 2.02↑ for the latter), whereas little difference could be found in those of XJ098.

As a marker of lipid peroxidation in plant cells, MDA was applied to evaluate the plant tolerance to the biotic or abiotic stresses [36]. During the PEG treatment, the concentration of MDA in XJ002 leaf tissues was consistently lower than that in XJ098 (Figure 2E). Interestingly, the *AET5Gv20023600* encoding lipoxygenase was observed to be down-regulated (log2FC 1.71↓) in the tolerant genotype (XJ002), while it was up-regulated (log2FC 1.20↑) in the sensitive genotype (XJ098). The lipoxygenase involved in leaf senescence of plants and abiotic stress, whose metabolic product contains reactive oxygen and oxygen radicals, can damage cell membranes [37]. Therefore, XJ002 might have a higher level of ability to alleviate the harmful effects of oxidative stress under drought stress than XJ098.

As an osmoprotectant, high levels of proline and WSS help the plant maintain low water potentials, protect proteins from possible damage and improve the activities of many enzymes [38,39]. In this study, the proline and WSS content of XJ002 was 1.61 and 2.10 fold higher compared with that of XJ098 at the highest accumulation content, respectively, indicating that XJ002 was more tolerant to osmotic stress than XJ098 (Figure 1). Accordingly, the *AET5Gv20478000* encoding trehalose-phosphatase was only up-regulated in XJ002 (log2FC 1.04↑), where it could be involved in the pathway of sucrose synthesis. Moreover, the *AET6Gv20051600* encoding sucrose 1-fructosyltransferase was shown to be more obviously up-regulated in XJ002 (log2FC 5.52↑) compared with that in XJ098 (log2FC 3.71↑). A previous study demonstrated that sucrose 1-fructosyltransferase initiated fructan biosynthesis and further contributed to plant resistance to environmental stresses [40]. Overall, the activation of osmotic regulation in the tolerant genotype suggests that the content of sucrose could play a key role in drought tolerance of *Ae. tauschii*.

### 3.2. Photosynthesis

One of the crucial effects of drought is the dramatic degradation in photosynthesis, which is believed to suppress leaf expansion and damage the photosynthetic system. Previous studies suggested that drought-induced stomatal closure limited the CO_2_ uptake by leaves, which was of great significance in drought stress tolerance [41,42]. Carbonic anhydrase is a zinc-containing metalloenzyme, and the specific relation between carbonic anhydrase and RuBisCO facilitates CO_2_ to interact with the latter to maintain its function [43]. Photosynthesis-related gene (*AET7Gv20237000*) encoding alpha carbonic anhydrase 7-like was shown to be more obviously up-regulated in XJ002 (FPKM 7.18; log2FC 2.99↑) compared with that in XJ098 (FPKM 1.54; log2FC 2.84↑). The results suggest that carbonic anhydrase may have a role in the drought stress response in the tolerant *Ae. tauschii.*

### 3.3. Effects of the Drought Response on Phytohormone Signaling

Even at very low concentrations, phytohormones are believed to regulate a variety of cellular processes in plants, and play critical roles and coordinate various pathways in signal transduction during an abiotic-stress response [44].

Abscisic acid (ABA) could regulate stomatal closure to minimize the water loss from plants under drought stress [45]. In this study, most members involved in synthesis and metabolism of ABA displayed obvious differences under control and drought stress, such as ZEP, ABA1, NCED, ABA2, AAO, and violaxanthin. However, the transcriptional changes were found to be similar between the two *Ae. tauschii* under drought stress. In addition, the expression level of *PYR/PYL*, *PP2C*, and *SnRK* involved in ABA signaling pathway was not found to be obviously different between the two *Ae. Tauschii* either. The results indicate that the signal transduction pathway of ABA might not be a major factor in causing the difference in drought tolerance between XJ002 and XJ098. Therefore, other phytohormones, like auxins, JA, BRs, GA, etc., might undertake important roles in drought stress. As multifunctional phytohormones, auxins are of great importance to plant growth and development [46]. The *AET5Gv20942800* encoding protein NRT1/PTR FAMILY, which might be involved in the basipetal transport of auxin and the regulation of the nitrate transporter NRT [47], was more up-regulated in the sensitive genotype XJ098 (log2FC 3.39↑) than the tolerant genotype XJ002 (log2FC 1.68↑). The result indicates that the *AET5Gv20942800* of XJ098 was more sensitive to drought stress than that of XJ002. Brassinosteroids (BRs) are a kind of steroid hormone, and play an important part in plant growth and development [48]. The *AET6Gv20782900* encoding protein EXORDIUM-like might play a factor in a brassinosteroid-dependent regulation of growth and development, and was differentially down-regulated in the two genotypes (XJ002, log2FC 3.26↓; XJ098, log2FC 5.63↓). Jasmonic acid (JA) is proposed to have great effects on stomatal closure during drought stress [49]. In the pathway of ‘alpha-linolenic acid metabolism (ko00592)’, the *AET7Gv20454000* encoding 12-oxophytodienoate reductase, which is involved in the biosynthesis of JA, was more up-regulated in the tolerant genotype (log2FC 3.57↑) compared with the sensitive genotype (log2FC 2.48↑). As tetracyclic diterpenoid carboxylic acids, gibberellins (GA) are assumed to have positive roles in seed germination, stem elongation, flower and fruit development and leaf expansion [50]. The *AET3Gv20908200* encoding gibberellin 20 oxidase 2-like was consistently up-regulated in both genotypes, but the expression levels of DEGs were higher in XJ098 (log2FC 2.42↑) than XJ002 (log2FC 1.05↑). The results indicate that differential regulation of genes associated with phytohormone signaling exists between XJ002 and XJ098 under drought stress.

### 3.4. Signal Transduction

Under drought stress, plant cells start to perceive and receive signals mediated by G protein coupled receptors (GPCR), phosphoinositides, or receptor-like protein kinases (RLKs), followed by the generation of second messengers (e.g., ROS and inositol phosphates). The second messengers could modulate the activity of protein kinases through controlling intracellular Ca^2+^ levels, which influences the protein phosphorylation cascade, finally activating transcription factors and regulating specific stress-regulated genes [51]. Signal transduction requires appropriate spatial and temporal coordination among all signaling molecules [52]. Activated by receptors/sensors such as protein tyrosine kinases, GPCR, and two-component histidine kinases, the mitogen-activated protein kinase (MAPK) pathways are responsible for the generation of compatible osmolytes and antioxidants. In this work, the *AET1Gv20883700* associated with MAPK was found to be significantly up-regulated in the tolerant genotype XJ002 (FPKM 14.39; log2FC 4.51↑) compared with little difference in the sensitive genotype XJ098 (FPKM 0.11; log2FC 2.16↑). As for the abiotic stress signaling, CDPKs are serine/threonine protein kinases with C-terminal calmodulin-like domains that directly bind Ca^2+^. A number of studies proved that CDPKs could be induced or activated by abiotic stresses, which might be involved in abiotic stress signaling [53]. The *AET1Gv20462200* associated with CDPK-related kinase 3-like was consistently up-regulated in both genotypes in this study, but the expression levels of DEGs were higher in XJ098 than XJ002. Protein kinases could phosphorylate and activate target proteins from signaling transduction in response to osmotic stress [54]. For instance, the *AET2Gv20867500*, associated with serine/threonine-protein kinase PBL19-like, was up-regulated 3.0-fold more in XJ002 than XJ098 under drought stress.

### 3.5. Transcriptional Factors

Transcription factors contribute greatly to regulating plant growth and development, and can also enhance the resistance to abiotic and biotic stresses through orchestrating regulatory networks [55]. Major plant TF families such as NAC, AP2, bZIP, MYB, and WRKY have been documented as important regulators in plant responses to various abiotic and biotic stresses [56,57,58]. In this study, the DEGs encoding WRKY and NAC families exhibited a more obvious difference between the two accessions. The DEGs (*AET1Gv20300200* encoding WRKY24, *AET1Gv20866400* encoding WRKY51, *AET7Gv20860400* encoding WRKY70) were more down-regulated in the tolerance genotype XJ002 compared with the sensitive genotype XJ098. As expected, the *AET3Gv20244700* encoding WRKY24 was shown to be more obviously up-regulated in the sensitive genotype XJ098, while little difference could be observed in the tolerance genotype XJ002. Actually, similar results have already been reported in previous studies, showing that TFs (WRKY24, WRKY51, WRKY70) could negatively regulate plant drought tolerance [59,60,61]. NAC proteins are plant-specific transcription factors that regulate the abiotic stress response and tolerance [62]. Nakashima et al. [63] reported that *OsNAC6* could positively regulate abiotic and biotic stress-responsive genes in rice. In this study, the *AET7Gv20524100* encoding NAC was observed to be obviously up-regulated in tolerance genotype XJ002 (log2FC 2.70↑) while little difference was detected in sensitive genotype XJ098.

## 4. Material and Method

### 4.1. Plant Materials

The 155 *Ae. tauschii* accessions used in this study are listed in Appendix A. The accessions marked as ‘XJ’ and ‘T’ represent those from Xinjiang and the Yellow River basin (Henan and Shannxi province), respectively. These materials all belong to L1 lineage and are preserved in Plant Germplasm Resources and Genetic Engineering Laboratory, Henan University.

### 4.2. Measurement of Coleoptile Length (CL)

The 20% (*w*/*v*) of PEG-6000 for simulated drought stress was performed in a seed germination experiment. Twenty seeds of each accession were surface-sterilized in a solution containing 70% (*v*/*v*) ethyl alcohol for 3 min and rinsed three times with distilled water. The seeds were then germinated and cultured with 10 mL 20 PEG-6000 solution in a plastic dish (10 cm × 10 cm × 5 cm) preset by 2 layers of filter paper. Afterward, all the dishes were transferred into an incubator at 25 °C for 2 days with a relative humidity of 70%. Ten seeds were selected randomly to measure the length of coleoptile with a ruler. Furthermore, the same procedure was performed with pure water as control treatment. Three replications were conducted as mentioned above. The index of drought tolerance is calculated as the CL ratio of treatment and control.

### 4.3. Drought Treatment of Ae. tauschii in Seedling Stage

Under the above mentioned culture condition with pure water, the 10-day old seedlings were transplanted into black plastic pots containing 1/2 Murashige and Skoog salts solution (pH 5.8) [64]. The seedlings at the trifoliate stage (about 15-day old) were divided into two groups: one control group and one drought-treated group exposed to 20% PEG-6000. The seedlings were treated for 3, 5, 8, 10 and 20 h in the physiological and biochemical test, respectively. In addition, the seedlings were treated for 8 h in the RNA-seq experiment. The treatment was performed with three replications in a completely randomized design.

### 4.4. Characterization of Water Loss Rate and Physiological Traits

Rate of water loss (RWL) was measured at the seedling 3-leaves stage, referring to the method of Dong et al. [65] with slight changes. The leaf samples were separated, weighed immediately (W_1_) and kept for 0 h, 0.5 h, 1 h, 2 h, 3 h, 4 h, 5 h and 6 h under ambient laboratory conditions at 25 °C, respectively reweighed (W_n_), oven-dried for 2 h at 50 °C, and weighed again (W_3_). From these weights, rate of water loss per unit leaf dry weight was calculated by the following equation:RWL= (W_1_−W_n_)/(W_1_−W_3_) ×100%(1)

Proline contents were determined by spectrophotometer, referring to the method of Bates et al. [66] with slight changes. The fresh leaves (0.2 g) were homogenized in 5 mL sulphosalicylic acid (3% in concentration). After centrifugation at 10,000× *g* for 5 min, 2 mL supernatant, 2 mL glacial acetic acid and 4 mL ninhydrin (2.5% in concentration) solution were uniformly mixed, followed by heating at 100 °C for 30 min, and then quickly cooled down to room temperature. Afterward, 4 mL toluene was added to the mixture and the organic phase was extracted for absorbance measurement at 520 nm. Proline concentration was determined using a calibration curve and expressed as μg proline/g.

Peroxidase (POD) was determined, referring to Zhang et al.’s work [67], by monitoring the changes of absorption at 470 nm accompanied with guaiacol oxidation. The reaction buffer solution was firstly prepared with 50 mM PBS (pH 6.0), 5 mM H_2_O_2_ and 10 mM guaiacol, which was then added to the sample solution to start the reaction. The activity was calculated from the change in absorbance at 470 nm for 1 min and expressed as AU/mg protein.
POD activity (AU/mg prot) = ∆A_470_ × 7133 ÷ Cpr(2)

Cpr: sample protein concentration, determined from BCA Protein Assay Kit (Signalway Antibody LLC, USA), according to the manufacturer’s protocol.

Polyphenol oxidase (PPO) activity was determined by detecting the increase in absorbance at 410 nm for catechol based on the method described by Yemenicioglu [68]. The reaction mixture contained 20 mM catechol and 1 mL enzyme solution. One unit of PPO activity was defined as the divergence in absorbance at 410 nm for 1 min and expressed as AU / mg protein.
PPO activity (AU/mg prot) = ∆A_410_ × 60 ÷ Cpr(3)

Cpr: sample protein concentration.

Malonadehyde (MDA) content was extracted and measured as described previously [69]. The fresh leaves (0.2 g) were firstly homogenized in 3 mL 10% trichloroacetic acid (TCA), and centrifuged at 10,000× *g* for 10 min. Then, 2 mL supernatant was added into 2 mL 0.6% thiobarbituric acid (TBA). The mixture was heated at 100 °C for 15 min, quickly cooled, and followed by centrifugation at 10,000× *g* for 10 min. The absorbance of the supernatant was collected at 450, 532, and 600 nm, respectively. The MDA content was calculated by the following equation:MDA (nmol/g) = 5 × (6.45 × (OD_532_−OD_600_) − 0.56 × OD_450_) ÷ W(4)

W: mass of leaf (g)

Water soluble sugar (WSS) content was measured by a spectrophotometer based on the method described by Pei et al. [70]. Absorbance of the supernatant was monitored at 620 nm. The WSS content was calculated by the following equation:WSS (μg/g) = ((∆A_620_−0.06) ÷ 3.39) ÷ W(5)

W: mass of leaf (g)

Relative electrolyte leakage reflects membrane damage, which was obtained through the method reported by Yan et al. [71]. The fresh leaves were washed three times with deionized water to remove surface-adhered electrolytes. The leaves (2 cm long strips) were percolated in a sealed centrifuge tube containing 10 ml deionized water for 12 h. Electrolyte leakage of the solution (L_t_) was determined at 25 °C. Samples were then incubated in boiling water for 30 min and the final electrolyte leakage (L_0_) was obtained after equilibration at 25 °C. Relative electrolyte leakage was defined by (L_t_/L_0_) × 100%. In each of the above experiments, three replicates were measured at each time point, respectively.

### 4.5. RNA Isolation and Sequencing

Total RNA was extracted using TRIzol reagent (Invitrogen Corp., Carlsbad, CA, USA), according to the manufacturer’s protocol. RNA quality was detected on 1% agarose gels. RNA purity was measured using a nanophotometer (Implen, Inc., Westlake Village, CA, USA). RNA concentration was checked by Qubit RNA Assay Kit in Qubit 2.0 Flurometer (Life Technologies, CA, USA). RNA integrity was assessed through the RNA Nano 6000 Assay Kit of the Bioanalyzer 2100 system (Agilent Technologies, CA, USA).

RNA-Seq library preparation and sequencing were carried out by Biomarker Technology Co., Ltd. (Beijing, China). Briefly, mRNA in the samples was enriched using oligo magnetic adsorption and the obtained RNA was fragmented into short pieces in a buffer. The cDNA was then synthesized utilizing these cleaved mRNA fragments as templates. Afterward, cDNA fragments with suitable lengths (300–500 bp) were obtained by agarose gel electrophoresis. Then PCR amplification was performed to enrich the purified cDNA template. Totally four libraries of both drought treated/untreated *Ae. tauschii* with contrasting drought tolerance were sequenced using an Illumina HiSeq™ 2000 (Biomarker Technologies Corporation, Beijing, China). For each sample, three biological replications were performed and further analyzed.

### 4.6. Mapping of Sequencing Reads and Quantification of Gene Expression

The rRNA contaminated from raw data was firstly removed by the Blast of rRNA database (https://www.arb-silva.de/), and the adapters, low-quality reads, and the reads containing poly-N were eliminated by the fastp program [72] with the parameters “-w 4 -q 20 -u 50” to obtain clean reads. The clean data were then deposited in the Short Read Archive database of NCBI with the accession number SRP154542. Next, the clean reads were processed and mapped to the *Aegilops tauschii* genome [22] utilizing the Hisat2 software (version 2.0.5) [73] with default parameters. The StringTie (version 2.0) was used to count the read numbers mapped to each gene [74]. The FPKM (Fragments Per Kilobase of exon model per Million mapped reads) of each gene was calculated based on its length and the mapped read numbers [75].

### 4.7. Expression and Enrichment Analysis of DEGs

The DEGs between control and stress samples were detected by the R procedure in Bioconductor package DESeq2 [29]. Meanwhile, the *q*-values were adjusted using Benjamini and Hochberg’s approach to control the false discovery rate [76]. The gene with *p*-value (*q*-value) (≤0.05) and an absolute value of log2 fold changes (|log2FC| ≥ 1) was considered to be differentially expressed.

Gene Ontology (GO) and Kyoto Encyclopedia of Genes and Genomes (KEGG) analysis were performed to identify the DEGs enriched in GO terms and metabolic pathways, respectively. For functional categorization and pathway visualization of DEGs, the Interproscan program [77] was applied to annotate GO and gene function. In addition, the metabolic pathways of DEGs were predicted through the KofamKOALA with default parameters [78]. The DEGs in the KEGG pathways and GO analysis were enriched by ClusterProfile R packages [30]. A corrected *p*-value (*q*-value) (≤ 0.05) was determined to be the threshold for significantly enriched GO terms and KEGG pathways. Transcription factors were determined using iTAk software [79].

### 4.8. Real-Time PCR Analysis

To validate the gene expression, real-time PCR was carried out using randomly selected genes. The cDNA synthesis was performed using iScript™ cDNA synthesis kit (Bio-Rad, Hercules, CA, USA). The qRT-PCR was performed in a 10 μL reaction volume using CFX96 system (Bio-Rad) and three biological replications were conducted for each reaction. The wheat actin gene was used as an internal reference and the quantitative primers were designed specifically to the gene of interest. The PCR process was set as follows: 30 s at 95 °C pre-denaturation, cycling 95 °C for 30 s, 60 °C for 30 s and 72 °C for 60 s for 40 cycles. Elongation was set at 72 °C for 5 min. Melt curve analysis was performed at 60–95 °C, with 0.5 °C increments for 5 s per step. The quantitative primers are listed in Appendix A.

### 4.9. Statistical Analysis

Coefficient of variation (C.V.) and t-test were calculated among the coleoptile length using Microsoft Excel 2013. Analysis of systematic clusters was performed through DPS software package [80].

## 5. Conclusions

In the present study, XJ002 was demonstrated to be more tolerant in drought stress compared with XJ098. Lower RWL, more osmoprotectants (such as proline and WSS), higher activity of antioxidase (POD and PPO), and fewer cell oxidative substances (MDA and relative electrolyte leakage) were found in the tolerance genotype XJ002 under simulated drought stress. Transcriptome analysis indicated that DEGs with obvious differences between the two accessions could be enriched in the corresponding GO and KEGG pathways related to drought stress. It also implied that the transcriptional activity might be important features responsible for drought tolerance. The above results will contribute to characterizing the most critical pathways associated with drought stress in *Ae. tauschii* and identifying individual genes involved in drought tolerance mechanisms.

## Figures and Tables

**Figure 1 ijms-21-03595-f001:**
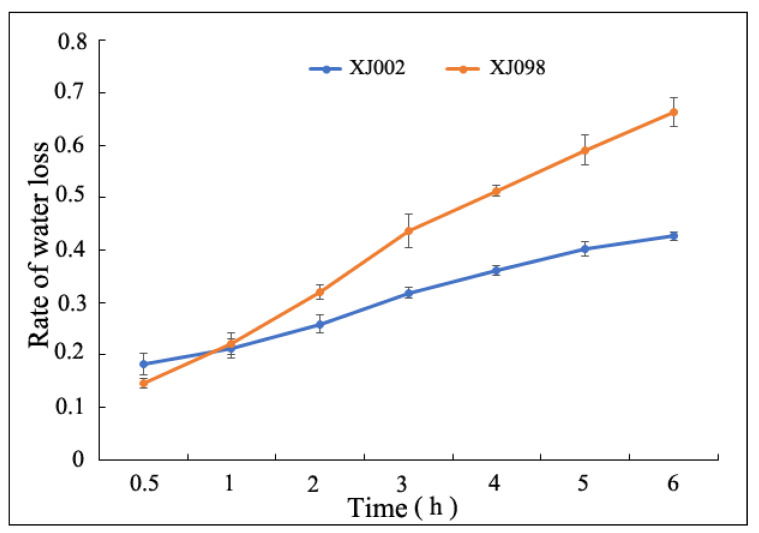
Rate of water loss from the two *Ae. tauschii* with contrasting drought tolerance. (Blue line: XJ002; Orange line: XJ098.).

**Figure 2 ijms-21-03595-f002:**
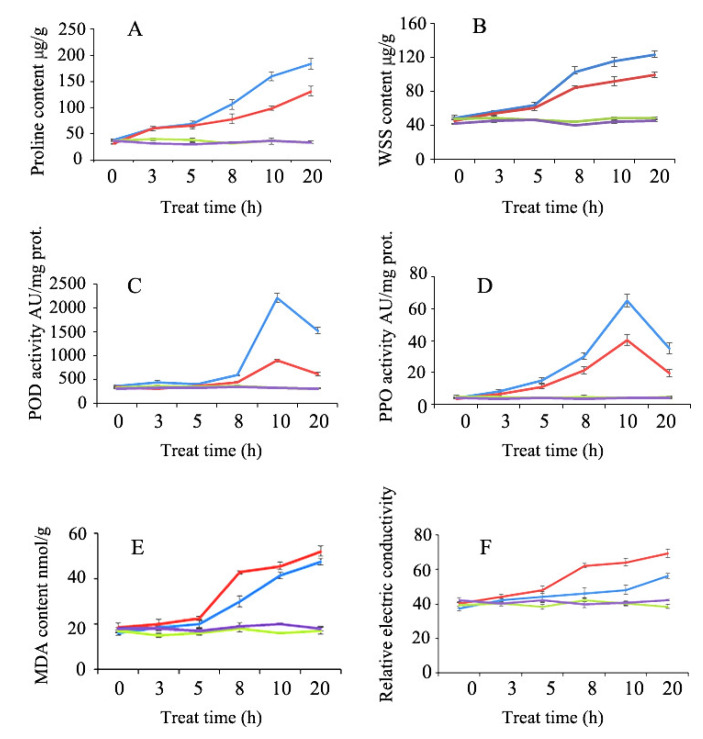
Physiological and biochemical characters of two *Ae. tauschii* (XJ002 and XJ098) with contrasting drought tolerance. (**A**): Proline content; (**B**): WSS content; (**C**): POD activity; (**D**): PPO activity; (**E**): MDA content; (**F**): Relative electrolyte conductivity. Blue line: XJ002_treatment; Red line: XJ098_treatment; Green line: XJ002_control; Purple line: XJ098_contorl.

**Figure 3 ijms-21-03595-f003:**
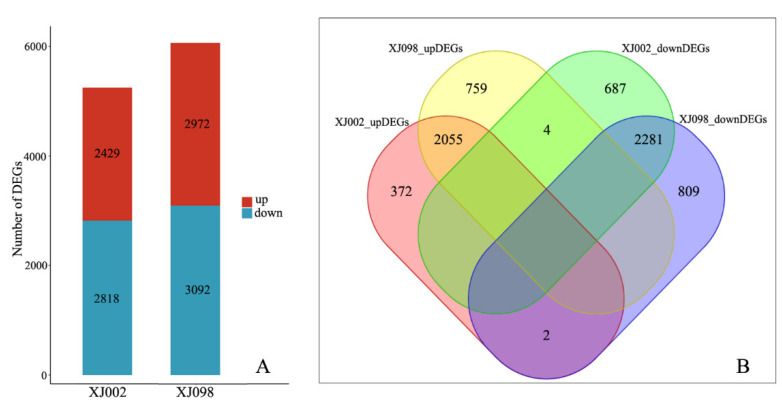
RNA-Seq analysis of two *Ae. tauschii* with contrasting drought tolerance. (**A**): Number of DEGs in both tolerant and sensitive genotype (red = up-regulated; blue = down-regulated). (**B**): Venn diagram of unique and common DEGs between the two genotypes.

**Figure 4 ijms-21-03595-f004:**
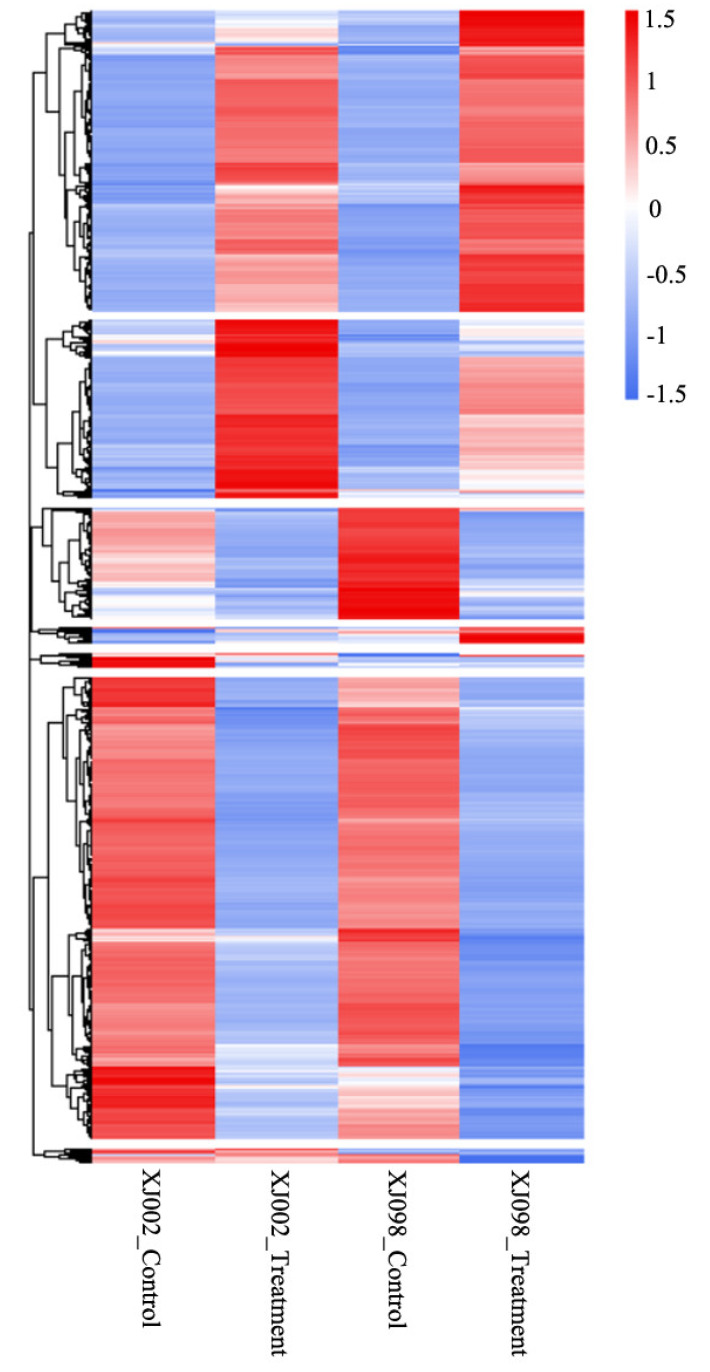
Hierarchical cluster analysis of 9184 DEGs with drought tolerance based on the log (FC) of gene expression. The color gradient from low (blue) to high (red) represents relative levels of gene expression. The numbers in the scale bar stand for the score of gene expression.

**Figure 5 ijms-21-03595-f005:**
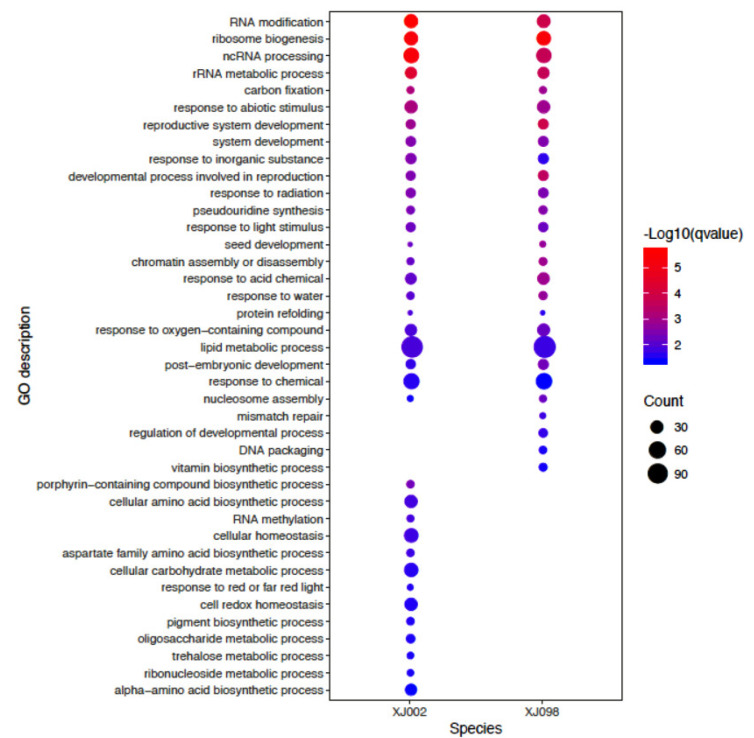
Categories and distribution of GO terms in the XJ002 and XJ098 under control and drought stress.

**Figure 6 ijms-21-03595-f006:**
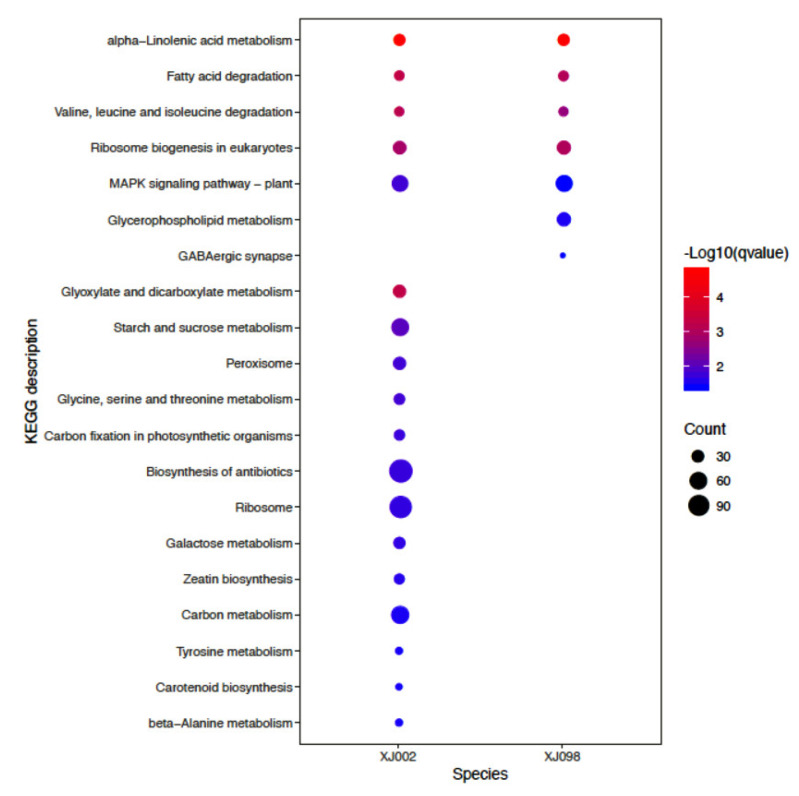
KEGG pathways prominently enriched DEGs in the XJ002 and XJ098 under control and drought stress.

**Table 1 ijms-21-03595-t001:** Summary of the sequence data from RNA sequencing.

Sample	Replication	Raw Reads	Clean Reads	Q20 (%)	Mapped Reads	Mapping Ratio (%)	Multiple Mapping Ratio (%)
XJ002_control	1	46,349,162	38,752,304	97.08	35,901,455	92.64	2.42
	2	49,217,242	41,052,492	97.07	38,004,210	92.57	2.52
	3	48,497,986	40,405,636	97.05	37,371,322	92.49	2.38
XJ002_treatment	1	41,805,526	34,966,454	97.08	32,598,302	93.23	3.13
	2	46,962,486	39,331,030	97.15	36,711,949	93.34	3.69
	3	48,556,172	40,506,772	97.15	37,687,700	93.04	3.51
XJ098_control	1	53,714,016	45,012,716	97.07	41,442,290	92.07	2.38
	2	49,159,574	41,084,578	97.11	38,190,927	92.96	2.49
	3	53,185,184	44,593,904	97.08	41,502,008	93.07	2.48
XJ098_ treatment	1	59,444,596	49,659,998	96.97	46,194,370	93.02	3.32
	2	49,422,820	41,393,532	97.11	38,237,463	92.38	2.91
	3	45,265,628	37,831,174	97.11	35,235,962	93.14	3.15

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
