# Peer review of "Comparative Transcriptome Analysis of Two Aegilops tauschii with Contrasting Drought Tolerance by RNA-Seq"

_ijms, 2020, doi:10.3390/ijms21103595_

Round 1

Reviewer 1 Report

Comments to the authors

This manuscript investigated transcriptomic profiles of Aegilops tauschii in response to drought stress. A tolerant and susceptible accessions were screened from 155 accessions based on plant length under drought stress. Some of physiological and biochemical parameters were studied and a tolerant accession accumulated much more osmolytes and increased ROS scavenging enzyme activities. RNA sequence analysis identified differentially regulated genes between the two accessions.

Major comment

One of my serious concerns is whether measurement of plant length can be a good indicator of stress tolerance in plants. For evaluation of agronomic performance, biomass production of vegetative organs is critically important. Please show the data of fresh weight and dry weight to fairly valid the degree of drought tolerance in these accessions.

Another concern is although the study focused on drought tolerance, there is no data demonstrated on water relations such as leaf water potential. This information is very important to know how plants tested suffered from drought stress.

Please make linkage between physiological evaluation of drought tolerance and transcriptome profiling. For example, a tolerant accession showed better physiological and biochemical parameters shown in Figure 1, however, gene expression profiles responsible for these parameters were never mentioned.

I think, to strengthen the manuscript, some additional assays should be tested to verify drought tolerance in XJ002. Numerous changes in gene expression were detected, however, evidence on how these gene products make superior tolerance to drought stress in a tolerant accession is lacking.

Reevaluate enzyme activity based on protein concentration. In Figure 1, plant weight was used for calculation of enzyme activity, however, weight is very changeable seriously affected by drought stress.

Minor comment

Use micro, but not “u” in y-axis of Figure 1

Explain how the expression data by qRT-PCR were normalized using actin.

Reviewer 2 Report

Figure 1 - there were no controls for no drought for these lines.  It is possible that biochemical characteristics measured vary with the circadian clock and do not show the differences in response to stress.  No details on how many plants were measured etc... in the results section or on the figure legend.  POD, PPO and other abbreviations should be spelled out in the text, not only in the materials and methods.  This analysis only included two lines - one potentially resistant and one susceptible to stress - and only for 20 hours of stress.  It is unclear from the results of the experiment if the differences reported are due to stress or due to individual variation of the lines, as well as it is unclear how common these responses are between the lines.  The analysis of more lines, as well as including no stress controls are important to make conclusions made in the paper. The quality control of the reads looks very good, but I would suggest adding a FPKM < 1 criteria in the beginning of the analysis to control for low expressing genes.  I would also suggest moving Figure 2 into supplemental figures.  I could not find Supplemental Figure 1, but the way, but it was probably my fault... The authors used 2-fold difference and a q<0.05 as an indication of the gene is differentially expressed or not.  They do not discuss how they treated genes that "almost" make the cutoff.  Since the number of replicates is relatively low, we would expect a significant number of genes to be close to the cutoff.  Since the authors compare the response in two genotypes, these "close to a cutoff in one genotype but not another" gene increase the number of reported differences between the genotypes, when in reality the responses are quite similar.  I would suggest implementing some cutoffs on calling the genes "definitely not DEG" and making comparisons using DEG and "not DEG" genes.   I would suggest selecting several genes with clear differences in patterns (highly overexpressed in one line and clearly not changing in expression level in another line) to show the differences as examples.   Figure 5 is confusing and the parameters of a statistical analysis of GO enrichment are not clear from the text.  This section does not clearly demonstrate which genes (GO groups / KEGG classes) show different expression patterns between two lines.  They look similar and the individual gene differences are not clear.  Figure 8 could be moved to supplemental materials.  I would suggest using qRT-PCR to analyze the selected genes with most interesting expression patterns in a larger group of lines to test whether identified expression patters are common among different lines. I would also suggest looking at promoter regions of the identified genes to see if there are special stress-related elements in them and whether there are differences in them between genes that are activated and genes that are not activated. Finally, the manuscript should be heavily edited for English - it has few typos and grammar errors, but the structure of the sentences and the use of many words and expressions make the manuscript more challenging to understand.

Author Response

Thanks very much for your valuable suggestions as well as recommendation on our paper. These constructive suggestions are of great significance to improve the quality of our manuscript. By taking the suggestions into consideration, the responses and corresponding revisions in the manuscript are listed below. All corrections made to the text have been highlighted in yellow type.

To Reviewer 2:

Figure 1 - there were no controls for no drought for these lines. It is possible that biochemical characteristics measured vary with the circadian clock and do not show the differences in response to stress.

Reply: Thanks for your kind reminding. Actually, the measurement of biochemical characteristics was conducted under simulated drought condition. With the extension of drought stress time, the response of the Ae. tauschii to drought becomes more and more obvious. At the end of stress, the decline of physiological traits indicates that the Ae. tauschii is on the threshold of death. Moreover, the original point (0 h) is untreated and could be used as a control for drought stress in this experiment.

No details on how many plants were measured etc... in the results section or on the figure legend.

Reply: Thanks for your advice. Five replicates of each Ae. tauschii accession were measured respectively at each time point. The details have been added in the Methods.

Addition: (Page 24, lines 517-518)

In each of the above experiments, 5 replicates were measured at each time point, respectively.

POD, PPO and other abbreviations should be spelled out in the text, not only in the materials and methods.

Reply: As the reviewer suggested, all the abbreviations have been spelled out in the revision.

This analysis only included two lines - one potentially resistant and one susceptible to stress - and only for 20 hours of stress. It is unclear from the results of the experiment if the differences reported are due to stress or due to individual variation of the lines, as well as it is unclear how common these responses are between the lines. The analysis of more lines, as well as including no stress controls are important to make conclusions made in the paper.

Reply: Thanks for your valuable suggestion. We strongly agree with your viewpoint. As reported in previous literatures, the response of coleoptile length (CL) to low water potentials demonstrates considerable genetic variability at very early seedling stage in wheat (Rebetzke et al., 2004; Dodig et al., 2015). In addition, coleoptile growth has been found positively related with grain yield under drought stress in wheat (Moud & Maghsoudi 2008). Similarly, CL was demonstrated to be a good parameter to distinguish drought-resistant types from drought sensitive wheat in the early seedling stage (Wang and Zou,1997). All the above researches establish CL as a desirable criterion, which was applied to preliminarily screen 155 Ae. tauschii accessions in this study. For the two obtained materials (XJ002 and XJ098) with obvious different CL, their physiological characteristics and rate of water loss (RWL) were further analyzed under osmotic stress and control condition (the RWL experiment has been supplemented in the revised manuscript). The results indicate that the CLs are positively related with drought stress. Finally, given the significant difference in drought resistance between the two Ae. tauschii accessions (XJ002 and XJ098), transcriptome analysis was carried out to identify the DEGs related to drought resistance.

References:

Rebetzke GJ, Richards RA, Sirault XRR, Morrison AD. Genetic analysis of coleoptiles length and diameter in wheat. Australian Journal of Agricultural Research, 2004, 55: 733–743.

Dodig D, Zorić M, Jović M, Kandic V. Wheat seedlings growth response to water deficiency and how it correlates with adult plant tolerance to drought. Journal of Agricultural Science, 2015, 153: 466–480.

Moud AM, Maghsoudi K. Application of coleoptile growth response method to differentiate osmoregulation capability of wheat (Triticum aestivum L.) cultivars. Research Journal of Agronomy, 2008, 2: 36–43.

Wang W, Zou Q. Studies on coleoptile length as criterion of appraising drought resistance in wheat. Acta Agronomica Sinica, 1997, 23: 459-467.

The quality control of the reads looks very good, but I would suggest adding a FPKM < 1 criteria in the beginning of the analysis to control for low expressing genes.

Reply: Thank you for your suggestion. According to your advice, the genes with FPKM> 1 in any one sample were retained before differential analysis, and the related results have been updated in the revision.

Origin:

A total of 6,624 genes were differentially expressed in tolerant genotype (XJ002), including 3,047 up-regulated and 3,577 down-regulated genes. In comparison, totally 7,456 DEGs were identified in sensitive genotype (XJ098), with 3,614 up-regulated and 3,842 down-regulated genes under drought stress conditions (Fig. 3A).

Revision:

A total of 5,401 genes were differentially expressed in draught-tolerant genotype (XJ002), with 2,429 up-regulated and 2,972 down-regulated. In comparison, totally 5,910 DEGs were identified in drought-sensitive genotype (XJ098), with 2,818 up-regulated and 3,092 down-regulated under drought stress conditions (Fig. 3A).

I would also suggest moving Figure 2 into supplemental figures.

Reply: As the reviewer suggested, Figure 2 has been moved into supplemental materials in the revision.

I could not find Supplemental Figure 1, but the way, but it was probably my fault...

Reply: As could be observed in the revision, all Supplemental Figures have been attached in the end of the manuscript.

The authors used 2-fold difference and a q<0.05 as an indication of the gene is differentially expressed or not. They do not discuss how they treated genes that "almost" make the cutoff.  Since the number of replicates is relatively low, we would expect a significant number of genes to be close to the cutoff.  Since the authors compare the response in two genotypes, these "close to a cutoff in one genotype but not another" gene increase the number of reported differences between the genotypes, when in reality the responses are quite similar.  I would suggest implementing some cutoffs on calling the genes "definitely not DEG" and making comparisons using DEG and "not DEG" genes. I would suggest selecting several genes with clear differences in patterns (highly overexpressed in one line and clearly not changing in expression level in another line) to show the differences as examples.

Reply: Thanks for your professional analysis and kind reminding. In the revised manuscript, DEGs with obvious difference between the two samples have been analyzed in the Result section.

Addition: (Pages11-12, Lines 229-256)

The AET5Gv20023600 encoding lipoxygenase, belonging to ‘alpha-Linolenic acid metabolism (ko00592)’ pathway, was observed down-regulated (log2FC 1.71↓) in tolerant genotype (XJ002), while up-regulated (log2FC 1.20↑) in sensitive genotype (XJ098). Only up-regulation could be found in XJ002 (log2FC 1.04↑) for the AET5Gv20478000 (encoding trehalose-phosphatase), which was enriched in the ‘starch and sucrose metabolism (ko00500)’ pathway. Similarly, the AET6Gv20051600 encoding sucrose 1-fructosyltransferase was observed to be more obviously up-regulated in XJ002 (log2FC 5.52↑) compared with that in XJ098 (log2FC 3.71↑). The AET1Gv20883700 encoding mitogen-activated protein kinase kinase kinase (MAPKKK), belonging to ‘MAPK signaling pathway (ko04016)’ pathway, was established more apt to be up-regulated in XJ002 (log2FC 4.51↑) compared with that in XJ098 (log2FC 2.16↑). Moreover, the AET2Gv20156600 encoding peroxidase exhibited more explicit tendency of up-regulation in XJ002 (log2FC 2.02↑) while little discrepancy could be found in the XJ098.

In addition, the DEGs encoding WRKY and NAC families between the two accessions exhibited more significant difference compared with those of the other transcription factors (TFs). The AET1Gv20300200 encoding WRKY24 was found to be obviously down-regulated in XJ002 (log2FC 1.42↓) with little difference in XJ098. While the AET3Gv20244700 encoding WRKY24 showed more prominent up-regulation in XJ098 (log2FC 1.82↑) with little difference in the other species. For the AET1Gv20866400 encoding WRKY51, more down-regulation could be observed in the tolerance genotype XJ002 (log2FC 4.15↓) than the sensitive genotype XJ098 (log2FC 2.71↓). Similarly, the AET7Gv20860400 encoding WRKY70 was detected down-regulation in tolerant genotype XJ002 (log2FC 2.76↓), with the adverse tendency in sensitive genotype XJ098 (log2FC 1.96↑). Moreover, the AET7Gv20524100 encoding NAC was observed to be up-regulated in XJ002 (log2FC 2.70↑) with little difference in XJ098.

Figure 5 is confusing and the parameters of a statistical analysis of GO enrichment are not clear from the text. This section does not clearly demonstrate which genes (GO groups / KEGG classes) show different expression patterns between two lines.  They look similar and the individual gene differences are not clear. 

Reply: As the reviewer indicated, the previous results were not clear enough to demonstrate the differences between the two samples. By establishing more stringent screening criteria, the DEGs for GO and KEGG have been updated in the revision.

Addition1: (Page 10, Lines 202-210)

Especially, the GO terms (GO: 0006629) in XJ002 and XJ098 enriched DEGs higher than 100, which could be indexed to lipid metabolic process. The gene ontologies related to drought stress could also be observed, including ‘dioxygenase activity’, ‘glucosyltransferase activity’, and ‘calcium ion binding’. Further, the enriched GO terms from XJ002 and XJ098 were compared on the basis of their biological process (Fig. 5). As a result, those related to ‘cellular carbohydrate metabolic process (GO: 0044262)’ and ‘cell redox homeostasis (GO: 0045454)’ were significantly enriched in the tolerant genotype. While in the sensitive genotype, gene ontologies correlated to ‘regulation of developmental process (GO: 0050793)’ were observed.

Addition2: (Pages 10-11, Lines 216-222)

It is noteworthy that 2 pathways, ‘alpha-Linolenic acid metabolism (ko00592)’ and ‘MAPK signaling pathway (ko04016)’ were found enriched in both of the two Ae. tauschii. Besides, 3 pathways were detected in the drought-tolerant genotype, including ‘starch and sucrose metabolism (ko00500)’, ‘peroxisome (ko04146)’, and ‘carbon fixation in photosynthetic organisms (ko00710)’. The pathway, ‘glycerophospholipid metabolism (ko00564)’, could only be obtained in the drought-sensitive genotype.

Figure 8 could be moved to supplemental materials.

Reply: According to the reviewer’s suggestion, Figure 8 has been moved to supplemental materials in the revision.

I would suggest using qRT-PCR to analyze the selected genes with most interesting expression patterns in a larger group of lines to test whether identified expression patters are common among different lines. I would also suggest looking at promoter regions of the identified genes to see if there are special stress-related elements in them and whether there are differences in them between genes that are activated and genes that are not activated.

Reply: Very good suggestions. The reviewer provides a very good idea for further exploring the interesting target genes in subsequent research. We have screened relatively reliable DEGs is this work. Next, we would analyze their intrinsic mechanism in drought stress. Thanks for the valuable advices again.

Finally, the manuscript should be heavily edited for English - it has few typos and grammar errors, but the structure of the sentences and the use of many words and expressions make the manuscript more challenging to understand.

Reply: Thanks for your suggestion. The whole manuscript has been carefully polished in the revision.

Round 2

Reviewer 1 Report

Comments to the authors

This is the revised manuscript for investigation of transcriptome responses of Aegilops tauschii in combination with some of physiological and biochemical analyses. I think the authors have made corrections as following this reviewer’s comments well, however, there are still serious concerns remained in the manuscript.

Enzyme activity must be evaluated based on protein amount used in the assay and I think this is a very basic part of biochemistry. Enzyme activity expresses how much target proteins are active in crude protein extracts, therefore, weight-based calculation is meaningless. Two examples using the same evaluation method (Zhang et al., 2007 and Yemenicioglu, 2002) were indicated, however, these papers also seem to have the same issue that is not acceptable for evaluation of scientifically reliable data. Especially, in case of drought stress research, enzymatic activity must be evaluated based on protein amount, because plant weight is quickly affected after water deficit treatment as newly demonstrated in the results of water loss in Fig. 1.

Method section for evaluation of enzyme activity, “expressed as unit g-1 protein”, must be revised.

Author Response

Reply: Thanks for your valuable suggestion. The enzyme activity based on protein concentration was reevaluated in the revision. In addition, the physiological characteristics for the two lines with contrasting drought tolerance was re-measured under simulated drought stress and control condition (Fig. 2). The physiological traits of XJ002 and XJ098 displayed few changes in the control condition. While, under the drought stress treatment, the physiological characteristics of the two lines exhibited obvious difference.

Reviewer 2 Report

The authors modified the manuscript according to my recommendations.  I still would like to see stronger controls to identify genes that are truly DE.  I agree that the two lines showed clear response to drought, I am just arguing that many of the genes identified as DE are differentially expressed not due to response to drought but for some other reason.  I understand that to fix this issue the whole experiment should be redone and it is impossible.  I would like to see somewhere in the manuscript the acknowledgement of the fact that stronger controls would allow to better differentiate between genes responsive to drought and genes that are differentially expressed due to other causes.

Author Response

Reply: Thanks for your suggestion. The related content has been added in the revision. In addition, based on your previous suggestion, the physiological characteristics for the two lines with contrasting drought tolerance was re-measured under simulated drought stress and control condition (Fig. 2). The physiological traits of XJ002 and XJ098 displayed few changes in the control condition. While, under the drought stress treatment, the physiological characteristics of the two lines exhibited obvious difference.

Addition: (Page 13, lines 281-282):

It should be noted that finer controls would contribute to better discrimination between DEGs responsive to drought and other causes.

Round 3

Reviewer 1 Report

I do not have particular comments on this version of the manuscript.